# The Influence of Blade Type and Feeding Force during Resin Bonded Dentin Specimen Preparation on the Microtensile Bond Strength Test

**DOI:** 10.3390/mi12040450

**Published:** 2021-04-16

**Authors:** Apinya Limvisitsakul, Suppason Thitthaweerat, Pisol Senawongse

**Affiliations:** Department of Operative Dentistry and Endodontics, Faculty of Dentistry, Mahidol University, Bangkok 10400, Thailand; apinya.lim@mahidol.ac.th (A.L.); suppason.thi@gmail.com (S.T.)

**Keywords:** microtensile testing, blade, feeding force, bond strength, specimen preparation

## Abstract

This paper presents the effect of blade type and feeding force during resin-bonded dentin specimen preparation on the microtensile bond strength (μTBS) test. Forty resin-bonded flat middle dentin specimens were divided into four groups. The specimens of each group were sectioned according to type of blade and feeding force as follows: fine grit/20 N, fine grit/40 N, medium grit/20 N, and medium grit/40 N to obtain resin-dentin sticks with a cross-sectional area of 1.0 mm^2^. Four sticks from the center of each tooth were subjected to the μTBS test. Five remaining sticks of each group were selected for surface topography observation under a scanning electron microscope (SEM). As a result, the bond strength of the medium-grit group was higher than that of the fine-grit group (*p* < 0.001), whereas the feeding force had no influence on bond strength values (*p* = 0.648). From the SEM, sticks prepared with the fine-grit blade showed a smoother surface integrity and fewer defects on the specimen edges in comparison with the sticks prepared with the medium-grit blade. The grit type of the blade is one of the considerable factors that may affect the bond strength and the surface integrity of resin-dentin specimens for microtensile testing.

## 1. Introduction

Currently, several adhesive products have been launched with long-standing attempts to improve the mechanical, chemical, and biological properties of several dental adhesive systems. Various laboratory studies have been used to examine and predict the clinical performance of these new products [1]. The ideal requirements of dental adhesive material are good retention, marginal sealing, and long-term durability. There are various testing methods proposed by ISO/TS 11405—testing of adhesion to tooth structure, including bond strength measurement tests, gap measurement tests, microleakage tests, and clinical usage tests [2].

Tensile bond strength testing is a common method used for evaluating adhesion performance via controlled tension force through the center of the bonded area until debonding occurs. It is mainly classified by the size of the cross-section of the bonded area into two types of macrotensile bond strength tests and microtensile bond strength tests [3]. The macrotensile bond strength test refers to a test with a bonded interface larger than 3 mm × 3 mm. A larger bonded interface usually results in a lower bond strength because of the nonuniform stress distribution at the interface. Thus, the nonuniform stress distribution leads to cohesive failure. To solve this problem, Sano and coworkers introduced the microtensile bond strength test with a reduction of the interface size down to 1 mm × 1 mm. They used microtensile testing to measure the ultimate tensile strength and modulus of elasticity of mineralized and demineralized dentin [4]. Therefore, the microtensile bond strength test tends to achieve higher bond strength and predominantly adhesive bond failure in comparison with the macrotensile bond strength test [3,5,6]. Because of the smaller size of the microtensile specimen, the test allows assessing the bond strength to the irregular surface with small bonded areas, such as cavity walls, intraradicular dentin, and interesting regions. In addition, it is possible to explore the resin/dentin interfaces by using either a scanning electron microscope (SEM) or a transmission electron microscope (TEM). Furthermore, the microtensile bond strength test can be used as intratooth testing for any testing material with a split-tooth design. These factors have led to a decrease in the number of teeth needed for the microtensile test, resulting in a greater advantage in controlling tooth variation [7,8]. Moreover, the microtensile method is also a useful technique to investigate the effect of water on adhesive durability because of the lower time requirement for water infiltration into the small microtensile specimens [9,10].

Due to no internationally standardized test protocol of the microtensile testing laboratory procedures, different parameters and techniques were used, including tooth specimen preparation, aging process, and environmental conditions, so it was difficult to compare the results of the bond strengths reported by experiments with varied parameters and techniques. In the cutting process of microspecimen preparation, a metal bond circular blade with diamond abrasive is commonly used. While the blade cuts and moves through the specimen, it creates friction between the cutting blade and the surface of the specimen. The friction force may induce increased temperature, intrinsic fragility, and microcracks at the blade/surface interface, which may be harmful to the specimen. These outcomes may have a negative effect on the microtensile bond strength and integrity of microspecimens [11,12,13,14,15]. The defects which occur in the bonded area make it easier to create a crack propagation during the force loading in microtensile testing procedure, so it might be the reason for the lower microtensile bond strength [16]. Moreover, the presence of the defects in the small size of microtensile specimens may result in the increase of encountered premature failure, which is the debonding or cracking of specimens during specimen preparation prior to bond strength testing [7,8]. The bond strength value for premature failure specimens has been reported in many ways, such as the zero, mean, and lowest bond value. These different premature failure management approaches may be the cause of the unreliable data that affect the validity of microtensile testing [2].

However, there are many parameters in the cutting process of specimen preparation that may determine the outcome of microtensile testing. Therefore, this study evaluated the effect of blade type and feeding force during specimen preparation on the microtensile bond strength and surface integrity of bonded dentin to resin composite specimens. The null hypothesis tested was that different types of blades and feeding forces did not affect the microtensile bond strength of bonded dentin to resin composite specimens.

## 2. Materials and Methods

Forty extracted, non-carious human third molars kept in 0.1% thymol solution at 4 °C were used in this study under the protocol approved by the Faculty of Dentistry/Faculty of Pharmacy, Mahidol University Institutional Review Board, Thailand. The storage solution of extracted teeth was changed to normal saline solution (0.9% NaCl solution) 24 h before use.

### 2.1. Tooth Preparation

All third human molars were sectioned horizontally at the coronal 1/3 of the crown to expose the middle dentin with a low-speed diamond saw (Isomet^TM^; Buehler, Evanston, IL, USA) with a cutting speed of 200 rpm under water lubricant. Then, the root 2 mm below Cementoenamel junction (CEJ) was embedded in self-cured acrylic resin parallel to the CEJ level. The exposed middle flat dentin was perpendicular to the long axis of the tooth and load direction. The cutting surface of each tooth was polished in a linear motion (10 cm stroking) by silicon carbide paper 600 grit under running water for 30 s each, creating a standardized smear layer. Then, the remaining dentin thickness of 1.5 mm was monitored using radiographic examination. The obtained sticks with peripheral enamel or with a remaining dentin thickness of less than 1.5 mm were excluded from the microtensile bond strength test.

### 2.2. Bonding Procedure and Resin Composite Placement

After smear layer preparation, all moist dentin surfaces were subjected to a 1-step self-etching adhesive system (Prime & Bond Universal™, Dentsply DeTrey, Konstanz, Germany) according to the manufacturer’s instructions at room temperature. Then, composite resin (Filtek^TM^ Z350XT shade A2, 3M ESPE, Saint Paul, MN, USA) was placed on the dentin surface with a 2 mm incremental layer until a height of 4 mm was obtained. An LED light curing unit (Bluephase N, Ivoclar Vivadent AG, Liechtenstein, Germany) was used for optimal polymerization of composite resin with 20 s per layer. The light intensity of the curing unit was monitored at >1000 mW/cm^2^ before use. All bonded specimens were stored in distilled water at 37 °C for 24 h.

### 2.3. Specimen Preparation

Forty specimens were randomly divided into 4 groups (10 teeth/group) according to the type of blade and feeding force as follows: fine grit/20 N, fine grit/40 N, medium grit/20 N, medium grit/40N. Both blade types (Pace Technologies, Tucson, AZ, USA) had the same 4-inch diameter. The fine grit refers to 10–20 microns of diamond abrasive particle size (600 grit). The medium grit refers to 60–70 microns of diamond abrasive particle size (220 grit).

The acrylic base of the bonded tooth was fixed to a precision automatic cutting-off and grinding machine (ACCUTOM 50; Struers Inc., Cleveland, OH, USA). The bonded teeth specimens were sectioned parallel to the long axis of the tooth with a cutting speed of 1000 rpm under water cooling. The cutting direction was starting from the composite resin and ran through the adhesive and dentin. The diamond blade was routinely dressed before cutting a new resin-dentin bonded specimen. Blade dressing was accomplished at low speed (200 rpm) with light loading force (0.25 N) until the entire blade surface passed through an alumina wafer blade dressing stick.

For cutting procedures, the specimen block was fixed to an automatic cutting machine. From the first direction of cut, 1.0 mm thick slabs were obtained. Then, the second direction of the cut was achieved by rotating the specimen 90 degrees to the first direction. Rectangular-shaped sticks with an approximately 1.0 mm^2^ cross-sectional area were obtained. The four resin-dentin sticks at the middle of each tooth specimen were collected for measurement of bond strength, and eight remaining sticks from peripheral areas were kept. The obtained sticks with peripheral enamel or with a remaining dentin thickness of less than 1.5 mm were excluded from the microtensile bond strength test. Specimens that failed during the cutting process or before being subjected to the microtensile bond strength test were recorded as pretesting failures. Forty sticks per group were assigned for the microtensile bond strength test. The specimen’s preparation processes are demonstrated in Figure 1. Then, five sticks from 80 remaining sticks from each group were randomly selected and used for observation of the surface characteristics (voids, microcracks, and any defects on the specimens) and integrity under a scanning electron microscope (JSM-6610 LV, JEOL Ltd., Tokyo, Japan) at magnifications of 80×, 300×, and 500×.

### 2.4. Microtensile Bond Strength Test

The dimension of the bonded interface was measured with a digital caliper (Mitutoyo Corp., Tokyo, Japan). The microtensile bond strength test was measured with a universal testing machine (Lloyd™ Testing Machine, Model LR 10K, Lloyd Instruments, Fareham Hanth, UK) at a cross-head speed of 1 mm/min. Each stick was attached to a microtensile jig with a cyanoacrylate adhesive glue (Model repair II blue; Dentsply Sankin, Otawara, Japan) keeping parallel position to the long axis of the device on a universal testing machine. The microtensile bond strengths in megapascals (MPa) were calculated and recorded.

### 2.5. Failure Mode Observation

Both tooth and resin composite sites of the fractured specimens were observed under SEM at a magnification of 80–200×. Failure modes were investigated and classified into 4 types: type 1: adhesive failure (75% to 100% failure occurred at interface of resin dentin bond); type 2: mixed failure (mixed with adhesive failure at the resin/dentin interface and cohesive failure in resin and/or dentin); type 3: cohesive failure in dentin (75% to 100% of the failure occurred in the underlying dentin); and type 4: cohesive failure in resin (75% to 100% of the failure occurred in the adhesive resin and/or overlying composite).

### 2.6. Surface Topography and Integrity Observation

Five sticks from each group were randomly chosen to examine the surface characteristics and specimen integrity. The surfaces of the specimens were cleaned with an ultrasonic cleaner in distilled water for 10 min and gently blotted and dried using paper towels. The cleaned sticks were mounted on aluminum stubs and coated with palladium using a sputter coater machine (SC7620 Sputter coater, Quorum Technologies Ltd., Lewes, UK). The sticks were observed under SEM at 80×, 300×, and 500× magnifications. Then, two external edges of each stick were determined and calculated for a defect score. The defect score, which was a sum score of the depth and frequency of defects along the external edge of the sticks, was analyzed according to Table 1. The proportion of the appearance of defects along the edge of the specimen was observed under SEM at 300× magnification by placing the bonding interface at the center of the image under SEM [18]. The measurement of the depth and proportion score of each external edge of the microspecimen is shown in Figure 2.

### 2.7. Observation of Surface Roughness of Cutting Blade

Surface roughness of both fine and medium-grit blade used in this experimental were determined at the area of 2 × 2 mm^2^ under the optical lens at the magnification of 10x with a profilometer (Alicona InfiniteFocus SL, Graz, Austria). The average height of selected area (Sa) was analyzed. Moreover, the five arithmetical averages of the roughness profile (Ra) were analyzed from five collected linear lines of each type of blade using Alicona IF-Measure Suite Version 5.1 software (Alicona InfiniteFocus SL, Graz, Austria). The mean Ra values of each cutting blade were calculated from five Ra values.

### 2.8. Statistical Analysis

The collected data were analyzed using SPSS version 18 (SPSS Inc., Chicago, IL, USA). The means of the bond strength of four resin-bonded dentin sticks of each tooth were calculated and used as representative bond strength data for each tooth specimen. The normal distribution and homogeneity of variances of microtensile bond strengths were verified with Kolmogorov–Smirnov test and Levene’s test. Then, the influence of the cutting speed and feeding force was further analyzed with two-way ANOVA. The level of significance was set at *p* = 0.05. The zero bond strength value for each premature failure stick specimen was used and included for statistical analysis. Furthermore, the defect scores and the failure patterns of both sides of the debonded specimens were analyzed with the nonparametric Kruskal–Wallis test, and the specific sample pairs were compared with the Mann–Whitney U test with a significance level of *p* = 0.05.

## 3. Results

### 3.1. Microtensile Bond Strength Test

The means and standard deviations of the microtensile bond strengths are summarized in Table 2, which also shows the prevalence of premature failures in each group. The statistical analysis demonstrated that the type of blade had a statistically significant effect on bond strength (*p* < 0.001), while the feeding force did not demonstrate a significant influence (*p* = 0.648). The highest mean bond strength was detected in the medium grit/40 N group (32.10 ± 12.38 MPa). The lowest mean bond strength was found in the fine grit/20N group (17.67 ± 3.81 MPa). Due to the lack of influence of the feeding force factor, the bond strengths of both the 20 N and 40 N feeding forces with the same blade type were combined and are shown in Table 3. The mean bond strengths and standard deviations of the fine-grit group and medium-grit group were 18.12 ± 5.22 and 31.15 ± 12.14 MPa, respectively. An independent T-test was further used to analyze a significant difference in the bond strengths between the fine-grit group and the medium-grit group at a 95% confidence interval. The bond strength of the medium-grit group was significantly higher than that of the fine-grit group (*p* < 0.001).

### 3.2. Failure Mode Observation

The failure mode analysis is shown in Figure 3. From the Mann–Whitney U test analysis at a 95% confidence interval, there were statistically significant differences in the failure patterns between two different blade type groups (*p* < 0.001). All debonded specimens prepared with the fine-grit blade group presented an adhesive failure pattern. The prepared specimens with medium-grit blades showed all failure types with a large number of adhesive failures. Representative debonded specimen pictures under SEM are shown in Figure 4 and Figure 5.

### 3.3. Surface Topography and Integrity Observation

According to surface topography observations, the number of defect scores of each group was counted and is shown in Figure 6. The prominent defect score of the fine-grit group was 2, while the defect score of the medium-grit group was prominently shown as 3. A number of defect scores were significantly different between the fine and medium-grit groups regarding Mann–Whitney U testing (*p* = 0.044). The prepared sticks with the fine-grit blade showed smoother surface integrity and fewer defects, wrenches, and microcracks on the specimen’s external edge than the prepared sticks with the medium-grit blade. Representative images of each determined score are presented in Figure 7.

### 3.4. Observation of Surface Roughness of Cutting Blade

According to surface roughness measurement of the cutting blade, the Sa value of medium-grit and fine-grit were 4.82 and 1.68 micrometers (µm), respectively. The mean Ra values of medium-grit was 3.65 ± 0.27 micrometers, while the mean Ra values of fine-grit was 0.72 ± 0.03 micrometers. The 3D images analyzed by Alicona IF-MeasureSuite Version 5.1 software are shown in Figure 8.

## 4. Discussion

According to the results of this study, the blade type was a key factor that affected the microtensile bond strength. Thus, the null hypothesis regarding the type of blade was rejected. On the other hand, the effect of feeding force on microtensile bond strength was not found. The null hypothesis regarding the feeding force was accepted.

Microtensile testing is a common method used for determining bond strength. Regarding the relatively small size of the specimen, this test can permit the measurement of regional bond strengths, which show higher bond strength and more adhesive failures than macrotensile bond strength testing. However, the specimen preparation process of microtensile testing is labor-intensive and technically demanding. The small size of the specimens increases the risk of breakage before testing, which is designated as premature failure [7].

The resin-bonded dentin specimen consists of four layers with different elastic moduli. The modulus of elasticity of the adhesive layer is the lowest when compared to the hybrid layer, resin composite, and dentin [19,20,21]. Therefore, the adhesive layer is the weakest layer within the microtensile specimen. This layer may display less resistance to harmful force or stress. In this study, various defects on the resin-bonded dentin specimens observed under the SEM were found in consistency with the previous reports [11,22].

The cutting process is one of many steps used in specimen preparation for microtensile testing that may influence resin composite and dentin adherence measurements [8,23,24]. In this step, a cutting wheel or blade with a diamond abrasive is commonly used in the microscale abrasive material removal process. By using a single-point cutting edge blade, a linear movement between the microscopic single-point cutting edge of each abrasive grain particle and specimen shears a small chip of material removal from the specimen, resulting in abrasive wear. During the cutting process, a certain specific energy on the cutting surface is found when the abrasive particle on the blade moves through the specimen. The specific energy, whether friction, surface fatigue, or small vibration, may generate stress and microcrack distributions within the specimens [25,26,27,28,29,30].

The effect of the difference in the abrasive size of the diamond blade used during specimen preparation of resin-bonded dentin specimens for microtensile testing on the microtensile bond strength was determined in this study. The bond strengths of specimens prepared with the medium-grit blade were significantly higher than those of specimens prepared with the fine-grit blade. Moreover, only adhesive failures were found when specimens were prepared with a fine-grit blade. The specimens prepared with a medium-grit blade exhibited adhesive failure, mixed failure, cohesive resin, and cohesive dentin. This may be affected by the surface characteristics of the cutting blade. A fine-grit blade typically contains many small abrasive particles. The small abrasive particles of fine grit are embedded into a binder material, thus holding the particles in position. With the small particle grain size, close spacing to the specimen may be expected by using this fine-grit blade. Thus, using a fine-grit blade may create more of a polishing action when compared with a medium-grit blade [15]. Thus, the specimens subjected to fine-grit cutting in this study exhibited fewer flaws on surface tomography observation than medium-grit cutting. By means of the relative integrity of the prepared specimens with a fine-grit blade, the microtensile force tended to affect only the weakest part of the specimens, which are the adhesive layers of the resin-bonded dentin specimens. This might cause the adhesive failures observed in the fine-grit cutting group. Therefore, the bond strengths of the fine-grit cutting group were lower than those of the medium-grit cutting group.

The close structure of small abrasive size and the polishing action of a fine-grit cutting blade may cause debris clogging on the cutting surface. This may lead to an increase in the grinding force and specific energy that may induce surface fatigue or microcracks within the adhesive layer of the microspecimens [15]. This could reduce the bond strength in the fine-grit cutting group. Conversely, the medium-grit blade demonstrates more efficiency than the fine-grit blade for resin-bonded dentin specimen preparation. The medium grit of diamond particles is able to remove material more quickly and reduce the clogging of debris [30]. Because of the high cutting efficiency of the medium-grit blade, the effects of the cutting procedure in specimen preparation on the microtensile bond strengths were less than those of the fine-grit blade [15]. Thus, the bond strength of the medium-grit cutting group was higher than that of the fine-grit cutting group. However, more specimen defects and fewer adhesive failures were found. The increase in defects might influence the fewer adhesive failure in this study. Therefore, the prominent failure was adhesive failure at 63%, with mixed failure, cohesive failure in resin, and cohesive failure in dentin accounting for 9%, 5%, and 3%, respectively. In addition, the bond strength values of the medium-grit cutting group obtained from this study were relatively comparable to previous studies [31,32].

Unfortunately, the effect of different feeding forces could not be distinguished by using the same blade type in this study. This might be because the use of 20 N feeding force and 40 N feeding force was not sufficiently different to create an impact on the bond strength of the specimens.

Due to the current development of many dental materials, the dentist’s clinical decision in selecting the suitable product must be based on evidence-based practice [33]. The laboratory bond strength information was often referred to as clinical performance of the dental adhesive. The reliability of the bond strengths of the microtensile testing reported in the literature were variable. This report showed that the blade’s type in the specimen preparation is one of the factors that affect the bond strength value. Therefore, the dentist has to be more concerned about the inconsistencies associated with the method of microtensile testing that may lead to misinformed interpretation about the clinical effectiveness of the tested products.

Furthermore, other factors were not considered and included in this study, such as the diameter of the cutting blade, the blade thickness, the rigidity of the blade, and the mechanical properties of the specimens. These factors may induce vibrations in the circular cutting blade and may influence a specimen result in different ways. Further studies should be considered to assess these factors.

## 5. Conclusions

Within the limitations of this study, the specimen preparation using a fine-grit cutting blade produced lower bond strengths and more adhesive failure than a medium-grit cutting blade. The feeding forces do not affect bond strength. Moreover, the defect-free microtensile specimen was not found after the specimen’s preparation process. In the fine-grit cutting blade group, the external edge of microtensile specimens showed fewer defects than in the medium-grit cutting blade group.

## Figures and Tables

**Figure 1 micromachines-12-00450-f001:**
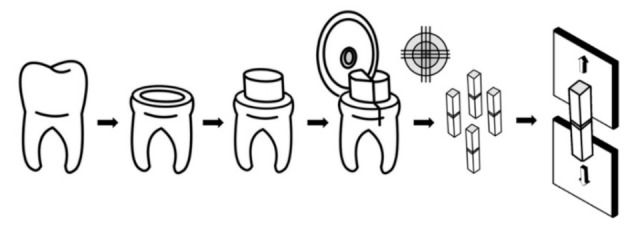
Diagram of the tooth preparation, resin composite placement, specimen preparation, and microtensile bond strength test (modified from Armstrong S et al., 2017) [17].

**Figure 2 micromachines-12-00450-f002:**
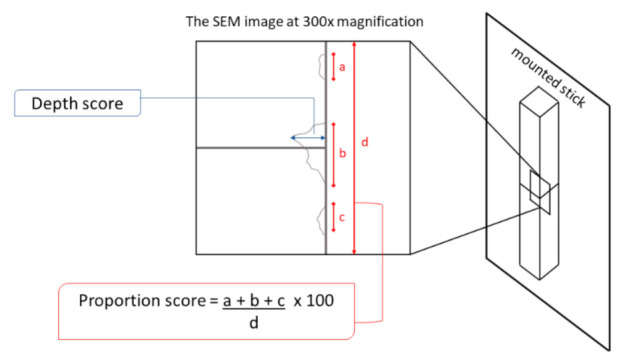
The measurement of the depth and proportion score of each external edge of the microspecimen [18].

**Figure 3 micromachines-12-00450-f003:**
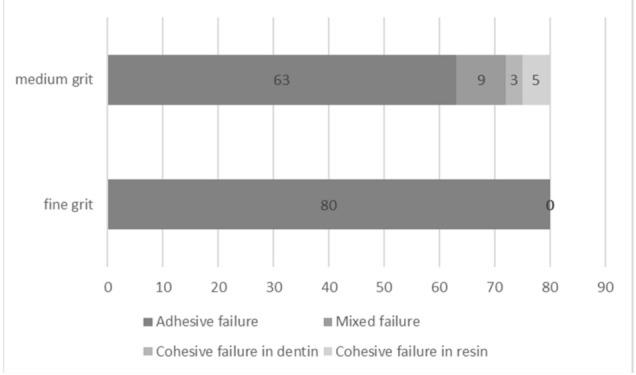
A number of failure modes of specimens prepared with either fine-grit or medium-grit blades are shown. The failure modes are classified into four types: adhesive, mixed, cohesive in dentin, and cohesive in resin.

**Figure 4 micromachines-12-00450-f004:**
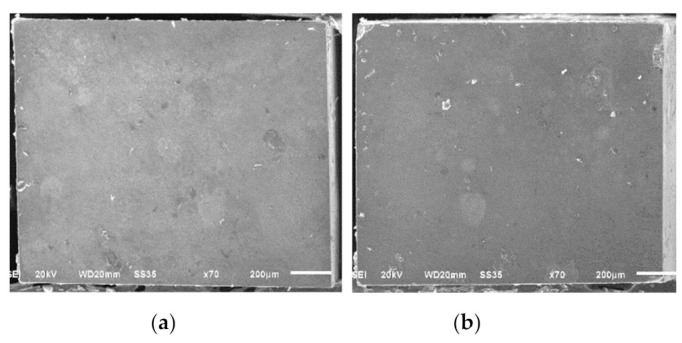
Representative SEM images of the surface of the debonded specimen: adhesive failure of a specimen from the fine-grit group. (**a**) Debonded specimens on resin composite fracture site of adhesive failure. (**b**) Debonded specimens on dentin fracture site of adhesive failure.

**Figure 5 micromachines-12-00450-f005:**
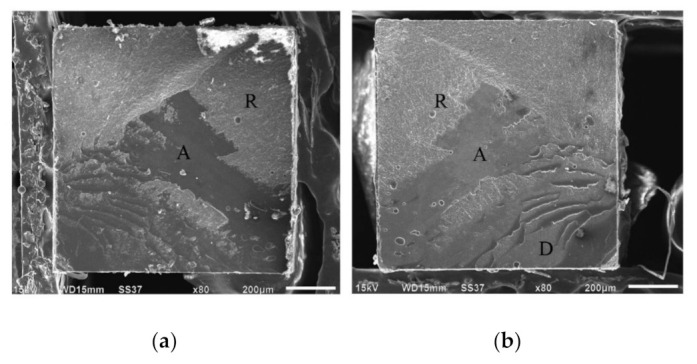
Representative SEM images of the surface of the debonded specimen: mixed failure of a specimen from the medium-grit group (A = adhesive, C = composite resin, D = dentin). (**a**) Debonded specimens on resin composite fracture site of mixed failure. (**b**) Debonded specimens on dentin fracture site of mixed failure.

**Figure 6 micromachines-12-00450-f006:**
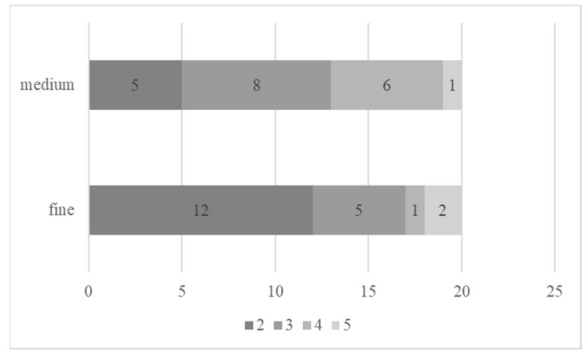
A number of sticks prepared with either fine-grit or medium-grit blades showing various external edge defects. Each color represents different defect scores from 2 to 5.

**Figure 7 micromachines-12-00450-f007:**
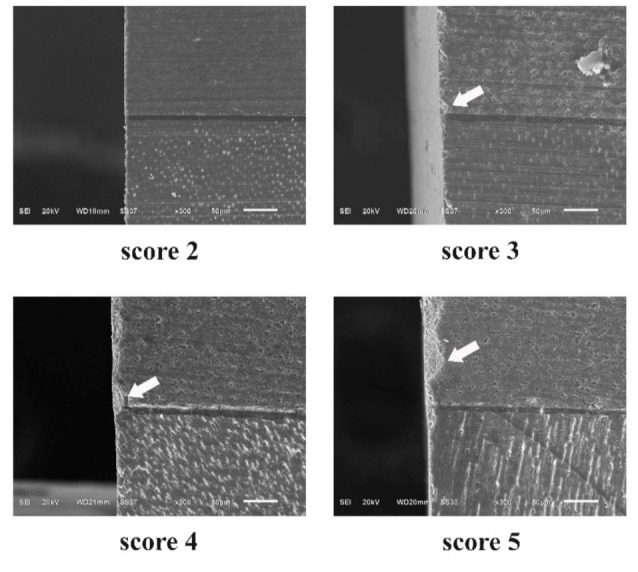
Representative SEM images of each defect score.

**Figure 8 micromachines-12-00450-f008:**
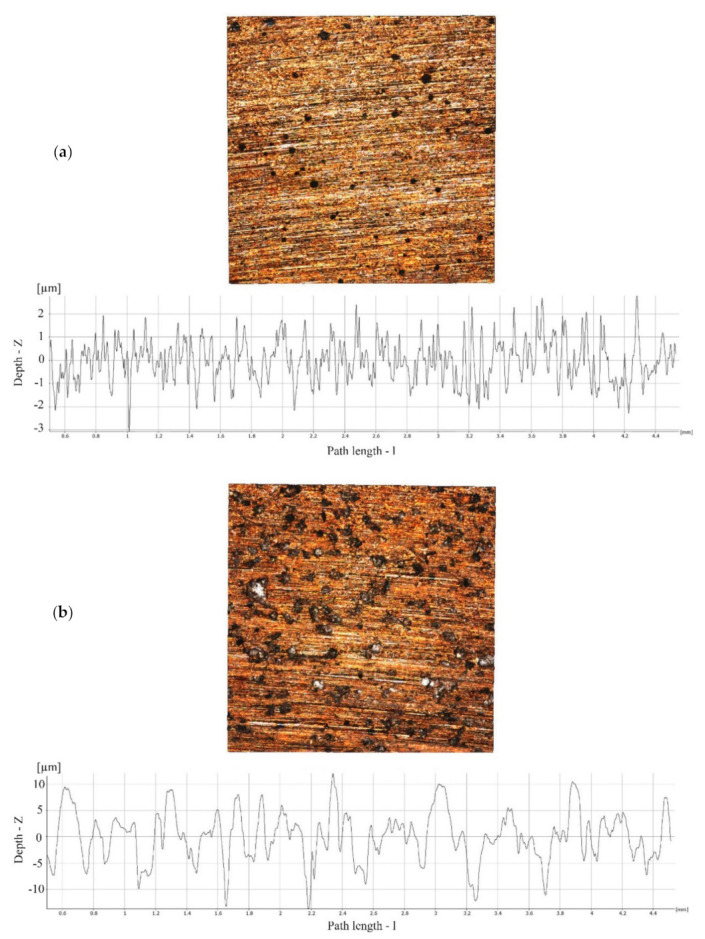
The representative 3D viewer pictures and 2D linear graphs showing a surface characteristic of fine-grit cutting blade (**a**) and medium-grit cutting blade (**b**).

**Table 1 micromachines-12-00450-t001:** Scoring of the depth of the external edge effect and the proportion of the appearance of the defect along the edge used for a defect score calculation in external edge defect assessment.

Score	The Depth of External Edge Defect	The Proportion of the Appearance of Defects Along the Edge of the Specimen at 300× Magnification
0	No defect appearance	No defect appearance
1	1–25 microns	1–25% of total image length
2	26–50 microns	26–50% of total image length
3	51–100 microns	51–75% of total image length
4	More than 100 microns	76–100% of total image length

**Table 2 micromachines-12-00450-t002:** Means and standard deviations (mean ± S.D.) of microtensile bond strength (MPa) of each group. The number of premature debonding specimens is presented in parentheses (*n* = 10/group).

Feeding Force (N)	Type of Blade	
	Fine Grit	Medium Grit
20	17.67 ± 3.81 (0)	30.21 ± 12.47 (1)
40	18.58 ± 6.53 (2)	32.10 ± 12.38 (0)

(N: newton).

**Table 3 micromachines-12-00450-t003:** Means and standard deviations (mean ± S.D.) of microtensile bond strength (MPa) of each group. The number of premature debonding specimens is presented in parentheses (*n* = 20/group).

Fine Grit	Medium Grit
18.12 ± 5.22 ^B^ (2)	31.15 ± 12.14 ^A^ (1)

^A, B^ Different letters indicate significant differences (*p* = 0.05). (N: newton).

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
