# Peer review of "The Influence of Blade Type and Feeding Force during Resin Bonded Dentin Specimen Preparation on the Microtensile Bond Strength Test"

_micromachines, 2021, doi:10.3390/mi12040450_

Round 1

Reviewer 1 Report

Esteemed collegue,

I have read your research paper and I liked the topic very much.

The paper is well written and the topic is actual but I would recommend some improvements.

It might be useful for the readers to now what is the practical impact of your research and also to include these pieces of information in your work.

That's why I advise you to revise the paper and focus more on practical influence of the results.

Please receive my warmest regards!

Author Response

Reviewer 1

We are thankful for the reviewer for the generous comments on the manuscript and we have edited the manuscript to address your concerns. We have added the clinical impact in the discussion section (in line322-328).

Due to the current development of many dental materials, the dentist's clinical decision in selecting the suitable product must be based on evidence-based practice [34]. The laboratory bond strength information was often referred to as clinical performance of the dental adhesive. The reliability of the bond strengths of the microtensile testing reported in the literature were variable. This report showed that the blade’s type in the specimen preparation is one of the factors that effect the bond strength value. Therefore, the dentist has to be more concern on the inconsistencies associated with the method of microtensile testing that may lead to misinformation interpreting about the clinical effectiveness of the tested products.

Also, we have already made a minor adjustment and also provided English edited certificate from AJE. This certificate was verified on the AJE website using the verification code 2E92-9D30-D393-A9CA-90C6.

Best regards,

Reviewer 2 Report

The article is well written and presented which conveys the idea for Resin Bonded Dentin Specimen Preparation.

Author Response

We are grateful to the reviewer for considering our work for publication. We have provided an English edited certificate from AJE. This certificate was  verified
on the AJE website using the verification code 2E92-9D30-D393-A9CA-90C6.

Reviewer 3 Report

The manuscript is of interest and the methods well described. I would suggest the authors to add to the discussion what is the clinical impact of their research and the potential implications in terms of everyday practice.

Author Response

Reviewer 3

We are thankful for the reviewer for the generous comments on the manuscript and we have edited the manuscript to address your concerns. We have added the clinical impact in the discussion section. (in line322-328)

Due to the current development of many dental materials, the dentist's clinical decision in selecting the suitable product must be based on evidence-based practice [34]. The laboratory bond strength information was often referred to as clinical performance of the dental adhesive. The reliability of the bond strengths of the microtensile testing reported in the literature were variable. This report showed that the blade’s type in the specimen preparation is one of the factors that effect the bond strength value. Therefore, the dentist has to be more concern on the inconsistencies associated with the method of microtensile testing that may lead to misinformation interpreting about the clinical effectiveness of the tested products.

Also, we have already made a minor adjustment and also provided English edited certificate from AJE. This certificate was verified on the AJE website using the verification code 2E92-9D30-D393-A9CA-90C6.

Best regards,

Reviewer 4 Report

The authors aimed to evaluate the effect of blade type and feeding force during specimen preparation on the microtensile bond strength and surface integrity of bonded dentin to resin composite specimens. The null hypothesis tested was that different types of blades and feeding forces did not affect the microtensile bond strength of bonded dentin to resin composite specimens.

The study is easy to follow and covers an interesting topic, but some  issues should be improved before publication. The manuscript needs moderate English change and grammar correction. Please also check typos thorough the text.

Discussion section: Will be useful to the reader to add some interesting literature related the issue, according to the evidence based-dentistry (please see and briefly discuss: DOI: 10.3390/ma8063268 DOI: 10.7150/ijms.4.174).

Conclusion Section: This paragraph required a general revision to eliminate redundant sentences and to add some "take-home message".

Author Response

Reviewer 4

We are thankful for the reviewer for the generous comments on the manuscript and we have edited the manuscript to address your concerns. We have already added the interesting points in line 274-276, including the take-home message in the discussion section according to your suggestion in line 322-328. Also, we have already revised the conclusion section.

Due to the current development of many dental materials, the dentist's clinical decision in selecting the suitable product must be based on evidence-based practice [34]. The laboratory bond strength information was often referred to as clinical performance of the dental adhesive. The reliability of the bond strengths of the microtensile testing reported in the literature were variable. This report showed that the blade’s type in the specimen preparation is one of the factors that effect the bond strength value. Therefore, the dentist has to be more concern on the inconsistencies associated with the method of microtensile testing that may lead to misinformation interpreting about the clinical effectiveness of the tested products.

Also, we have already made a minor adjustment and also provided English edited certificate from AJE. This certificate was verified on the AJE website using the verification code 2E92-9D30-D393-A9CA-90C6.

Best regards,